# Delta neutrophil index for predicting mortality in trauma patients who underwent emergent abdominal surgery: A case controlled study

Hui-Jae Bang[1], Kwangmin Kim[2], Hongjin Shim[1,3], Seongyup Kim[1,3], Pil Young Jung[1,3], Young Un Choi[1,3], Keum Seok Bae[1,3], Ik Yong Kim[1], Ji Young Jang[4]*

1 Department of Surgery, Yonsei University Wonju College of Medicine, Wonju, South Korea, 2 Department of Surgery, Saidabad Clinic, Dhaka, Bangladesh, 3 Regional Trauma Center, Wonju Severance Christian Hospital, Wonju, South Korea, 4 Department of Surgery, Trauma Center, National Health Insurance Service Ilsan Hospital, Goyang-si, South Korea

* drjangjiyoung@gmail.com, jyjang@hanmail.net

**Data Availability Statement:** All relevant data are within the manuscript and its Supporting Information files.

## Abstract

### Background

Delta neutrophil index (DNI) can be used as a biomarker for infection to predict patient outcomes. We aimed to investigate the relationship between DNI and clinical outcomes in trauma patients who underwent abdominal surgery.

### Materials and methods

We retrospectively analyzed injured patients who underwent emergent abdominal surgery in the regional trauma center of Wonju Severance Christian Hospital between March 2016 and May 2018. Patient characteristics, operation type, preoperative and postoperative laboratory findings, and clinical outcomes were evaluated. Logistic regression analysis was performed for risk factors associated with mortality.

### Results

Overall, 169 patients (mean age, 53.8 years; 66.3% male) were enrolled in this study, of which 19 (11.2%) died. The median injury severity score (ISS) was 12. The non-survivors had a significantly higher ISS [25(9–50) vs. 10(1–50), p<0.001] and serum lactate level (9.00±4.10 vs. 3.04±2.23, p<0.001) and more frequent shock (63.2% vs 23.3%, p<0.001) and solid organ injury (52.6% vs. 25.3%, p = 0.013) than the survivors. There were significant differences in postoperative DNI between the two groups (p<0.009 immediate postoperation, p = 0.001 on postoperative day 1 [POD1], and p = 0.013 on POD2). Logistic regression analysis showed that the independent factors associated with mortality were postoperative lactate level (odds ratio [OR] 1.926, 95% confidence interval [CI] 1.101–3.089, p = 0.007), postoperative sequential organ failure assessment score (OR 1.593, 95% CI 1.160–2.187, p = 0.004), and DNI on POD1 (OR 1.118, 95% CI 1.028–1.215, p = 0.009). The receiver operating characteristics curve demonstrated that the area under the curve of DNI on POD1 was 0.887 (cut-off level: 7.1%, sensitivity 85.7%, and specificity 84.4%).

**Funding:** This research received no specific grants from funding agencies in the public, commercial, or not-for-profit sectors.

**Competing interests:** An author [K.K] works as a volunteer at the Saidabad clinic, a commercial clinic. The funder did not provide support in the form of salary for author [K.K], and did not have any additional role in the study design, data collection and analysis, decision to publish, or preparation of the manuscript. This does not alter our adherence to PLOS ONE polices on sharing data and materials. All other authors have no competing interest. The authors have declared that no competing interests exist.

**Abbreviations:** DNI, delta neutrophil index; ISS, injury severity score; POD, postoperative day; OR, odds ratio; CI, confidence interval; SOFA, sequential organ failure assessment; ROC, receiver operating characteristics; AUC, area under the curve; WBC, white blood cell; CRP, C-reactive protein; IG, immature granulocyte; WSCH, Wonju Severance Christian Hospital; AIS, abbreviated injury scale; GI, gastrointestinal; ICU, intensive care unit; MPO, Myeloperoxidase; ER, emergency room; SIRS, systematic inflammatory response syndrome; DAMPs, damage-associated molecular patterns; MODS, multiple organ dysfunction syndrome.

## Conclusions

Postoperative DNI may be a useful biomarker to predict mortality in trauma patients who underwent emergent abdominal surgery.

## 1. Introduction

The current biomarkers for diagnosis of sepsis or infections include white blood cell (WBC) count, lactic acid, procalcitonin, and C-reactive protein (CRP) [1–3]. The release of immature neutrophils into the bloodstream during infection or sepsis leads to an elevation of the immature/total granulocyte ratio which is defined as neutrophil 'left-shift'. This granulocytic 'left-shift' or increase in immature granulocyte (IG) rate is commonly used as a diagnostic marker of infection or sepsis in the clinical setting. However, it is difficult to accurately measure IG using a microscopic examination of blood smears, and its diagnostic value remains controversial [4, 5]. Technological advances in an automated cell analyzer have enabled the acquisition of the delta neutrophil index (DNI) using leukocyte differentials obtained from two independent channels–the myeloperoxidase channel and the lobularity/nuclear density channel. The DNI is calculated as the difference between leukocyte differentials measured in these two channels, which reflects the proportion of circulating IG [6]. Several studies reported that DNI was associated with disease severities of sepsis or septic shock and mortality in patients with various infectious conditions such as bacteremia, pneumonia, and peritonitis [7–11]. Moreover, recent studies showed that DNI was associated with the severity and prognosis of non-infectious inflammation-related diseases, such as acute myocardial infarction, pulmonary embolism, upper gastrointestinal hemorrhage, and cardiac arrest [12–15]. However, there are few studies about the use of DNI in trauma patients. Therefore, the aim of this study was to evaluate the usefulness of DNI as a predictor of mortality in trauma patients who underwent emergent abdominal surgery.

## 2. Patients and methods

### 2.1 Patient selection and data collection

The study was approved by the institutional review board of Wonju Severance Christian Hospital (IRB no. CR319077). All data were fully anonymized before access and IRB waived the requirement for informed consent. Among 6291 injured patients who were admitted in the regional trauma center of a tertiary university hospital between March 2016 and May 2018, 173 patients who underwent emergent abdominal surgery were enrolled in this study. After exclusion of four patients who died within six hours of admission, the final study population was 169 (Fig 1).

The primary end-point was to evaluate the effectiveness of DNI to predict postoperative mortality in injured patients who underwent emergent abdominal surgery. The secondary end-point was to compare DNI with other biomarkers for prediction of mortality and to access the cut-off level of DNI. The demographic and clinical characteristics of the patients such as age, sex, injury mechanism, injury severity score (ISS), associated injury (abbreviated injury scale, AIS ≥3), initial shock, diagnosis, gastrointestinal (GI) perforation, solid organ injury, serum lactate, and sequential organ failure assessment (SOFA) score on intensive care unit (ICU) admission were retrospectively reviewed.

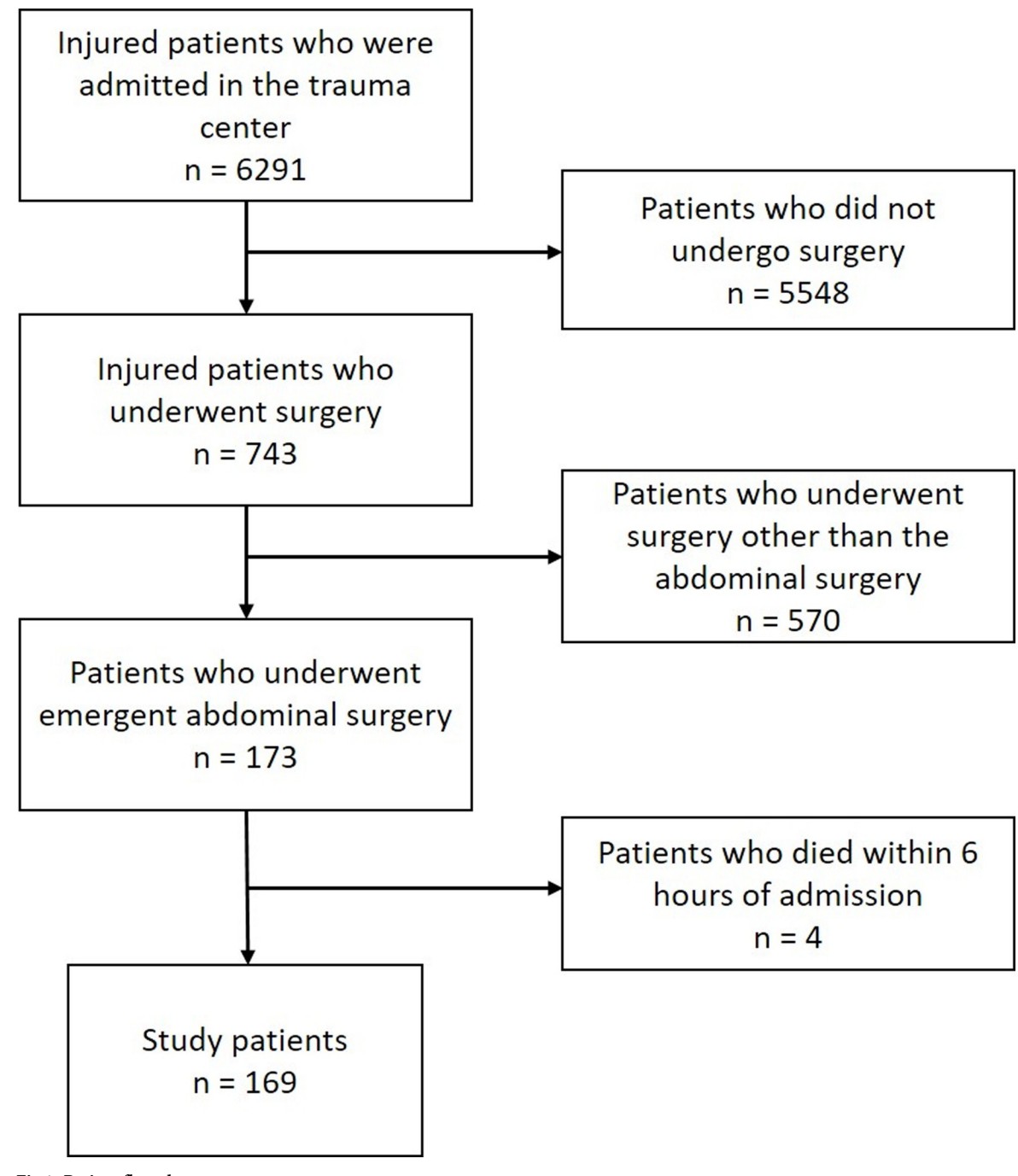

**Fig 1. Patient flow chart.**

## 2.2 DNI and laboratory tests

We reviewed the results of laboratory tests such as DNI, WBC count, and C-reactive protein (CRP), at four time points (initially in the emergency room [ER], immediate postoperative, postoperative day (POD) 1, POD2). In our institution, DNI is presented along with complete blood count tests without extra charge. A specific type of automatic cell analyzer (ADVIA 120/212; Siemens, Tarrytown, NY, USA) was used for the calculation of DNI. This flow cytometry-

based hematologic analyzer uses two independent WBC counting methods–a myeloperoxidase (MPO) channel and a lobularity/nuclear density channel. DNI value was calculated using the following formula: DNI = (leukocyte subfraction assayed using the MPO channel of a cyto-chemical reaction)–(leukocyte subfraction assayed using the nuclear lobularity channel based on reflected light beam measurements) [6].

## 2.3 Statistical analysis

Continuous variables were presented as the mean±standard deviation or the median values (ranges), and comparative analysis was performed using a Student's t-test. Categorical variables were analyzed by the Chi-square test and Fisher's exact test. To identify the independent risk factors for mortality, a multivariate analysis was performed using logistic regression. A receiver operating characteristics (ROC) curve was constructed, and the Youden Index method was used to find the optimal cut-off values for lactate, DNI, and SOFA score to predict mortality. All statistical analyses were performed using SPSS 20.0 (IBM, Armonk, NY, USA). Statistical significance was accepted for p <0.05.

## 3. Results

### 3.1 Baseline clinical characteristics

The study enrolled 169 consecutive injured patients who underwent emergent abdominal surgery during the study period (27 months). The mean age was 53.8±17.1 years, and 112 (66.3%) patients were men. Most common injury mechanism was road traffic collision, and 57 patients (33.7%) had associated injuries (AIS ≥3). The ISS was greater than 15 in 78 (46.2%) patients, and 47 (27.8%) patients initially had shock status. The mean level of serum lactate was 3.73 ±3.14 mmol/L, and the median SOFA score at the time of ICU admission was 4 (0–18). Nineteen (11.2%) patients died, and the most common cause of death was sepsis, followed by hemorrhage and multiple organ failure. Thirteen patients (68.4%) died within 7 days, and 4 patients (26.3%) died between 7 and 28 days. The other one died 28 days later (Table 1). The most common injury site was the small bowel (40.8%) followed by liver (11.8%), spleen (9.5%), and abdominal wall (5.9%). GI perforations were identified in 62 (36.7%) patients (Table 2).

### 3.2 Comparison between survivors and non-survivors

ISS [25(9–50) vs 10(1–50), p <0.001] and SOFA score [9.5 (6–18) vs 3 (0–15), p <0.001] on ICU admission were significantly higher in non-survivors than in survivors. Moreover, non-survivors had significantly more frequent associated injury (63.2 vs. 30%, p = 0.004), shock in ER (63.2 vs. 23.3%, p <0.001), and solid organ injury (52.6 vs. 25.3%, p = 0.013) than survivors. Serum lactate was significantly higher in non-survivors than in survivors on initial measurement in ER (9.00±4.10 mmol/L vs. 3.04±2.23 mmol/L, p <0.001) and immediate post-operation (7.64±3.54 mmol/L vs. 2.82±1.73 mmol/L, p <0.001) (Table 3).

WBC counts were significantly higher in survivors than in non-survivors on immediate post-operation (11096±5091 vs. 7161±5139, p = 0.002), POD1 (9846±3620 vs. 7017±3949, p = 0.005), and POD2 (8695±3080 vs. 5561±3972, p = 0.001) [Fig 2A]. In addition, CRP was significantly higher in survivors than in non-survivors on POD1 [10.40 mg/dL (0–35.80) vs. 2.80 mg/dL (0–17.30), p = 0.014] and POD 2 [15.20 mg/dL (0–35.10) vs. 7.32 mg/dL (0–32.10), p = 0.013] [Fig 2B]. DNI in non-survivor was significantly higher than in survivors on immediate post-operation [9.0 (2.3–48.8) vs. 3.3% (0–41.2), p = 0.009], POD1 [17.0 (1.4–57) vs. 1.5% (0–35.8), p = 0.001], and POD 2 [27.9 (0–62.1) vs. 0.8% (0–52.9), p = 0.004] [Fig 2C].

**Table 1. Patient characteristics.**

| Variable | N = 169 (%) |
|---|---|
| Age | 53.8±17.1 |
| Sex (male) | 112 (66.3) |
| Injury severity score | 13 (1–50) |
| Injury severity score > 15 | 78 (46.2) |
| Associated injury (AIS ≥3) | 57 (33.7) |
| Injury mechanism | |
| Road traffic collision | 104 (61.5) |
| Penetrating trauma | 31 (18.3) |
| Hit & crush | 19 (11.2) |
| Fall | 6 (3.6) |
| Slip down | 5 (3.0) |
| Others | 4 (2.4) |
| Initial shock | 47 (27.8) |
| Initial WBC | 12064±5904 |
| Initial DNI (%) | 0.6 (0–52.8) |
| Initial CRP (mg/dL) | 0.29 (0–30.10) |
| Initial serum lactate | 3.73±3.14 |
| Postoperative WBC | 10674±5226 |
| Postoperative DNI | 3.6 (0–48.8) |
| Postoperative CRP | 0.29 (0–26.40) |
| Postoperative lactate | 3.43±2.60 |
| SOFA score on ICU admission | 4 (0–18) |
| Mortality | 19 (11.2) |
| Sepsis | 7 (36.8) |
| Hemorrhage | 6 (31.6) |
| MODS | 2 (10.5) |
| Others | 4 (21.1) |

AIS, abbreviated injury scale; WBC, white blood cell; DNI, delta neutrophil index; CRP, C-reactive protein; SOFA, sequential organ failure assessment; MODS, multiple organ dysfunction syndrome.

## 3.3 Independent risk factors for mortality in critically ill and injured patients who underwent emergent abdominal surgery

The logistic regression model using variables that were noted in the univariate analysis showed that immediate postoperative lactate [odds ratio (OR) 1.926 [95% confidence interval (CI) 1.201–3.089], p = 0.007], SOFA score on ICU admission [OR 1.593 (95% CI 1.160–2.187), p = 0.004], and DNI on POD1 [OR 1.118 (95% CI 1.028–1.215), p = 0.009] were independent risk factors associated with mortality (Table 4).

## 3.4 Performance of DNI and other laboratory markers in critically ill and injured patients who underwent emergent abdominal surgery

When the ROC curves of the postoperative lactate, SOFA score on ICU admission, and DNI on POD1 were conducted to predict mortality, area under curve (AUC) of the immediate postoperative lactate, SOFA score on ICU admission, and DNI on POD1 were 0.874 (95% CI, 0.773–0.975, p <0.001), 0.941 (95% CI, 0.898–0.984, p <0.001), and 0.887 (95% CI 0.798–0.976, p <0.001), respectively. The optimal cut-off points for the postoperative lactate, SOFA

**Table 2. Patient diagnosis.**

| Diagnosis | N = 169 |
|---|---|
| Small bowel injury | 69 (40.8%) |
| Liver injury | 20 (11.8%) |
| Colorectal injury | 18 (10.7%) |
| Spleen injury | 16 (9.5%) |
| Abdominal wall injury | 10 (5.9%) |
| Major vascular injury | 9 (5.3%) |
| Pancreatic injury | 5 (3.0%) |
| Stomach injury | 4 (2.4%) |
| Other | 12 (7.1%) |
| Multi-organ injury | 6 (3.6%) |
| GI perforation | 62 (36.7%) |

GI, gastrointestinal.

Other; 5 omental injuries, 3 retroperitoneal hemorrhages, 1 gallbladder injury, 1 teratoma rupture, 1 renal injury, 1 none.

score on the ICU admission, and DNI on the POD1 were 5.105 mmol/L (sensitivity: 71.4%, specificity: 92.6%), 6.5 (sensitivity: 92.9%, specificity: 84.4%), and 7.1% (sensitivity: 85.7%, specificity: 84.4%), respectively (Fig 3).

The area under the ROC curve was 0.887 (95% confidence interval, 0.798–0.976) for the DNI (POD1).

- DNI POD1 (AUC = 0.887) (95% CI 0.798–0.976, $p < 0.001$)

  - cut-off level: 7.1%

  - sensitivity: 85.7%, specificity: 84.4%

- PostOP Lactate (AUC = 0.874) (95% CI, 0.773–0.975, $p < 0.001$)

  - cut-off level: 5.105

  - sensitivity: 71.4%, specificity: 92.6%

- PostOP SOFA (AUC = 0.941) (95% CI, 0.898–0.984, $p < 0.001$)

  - cut-off level: 6.5

  - sensitivity: 92.9%, specificity: 84.4%

## 4. Discussion

This study showed that DNI on the POD1 was an independent risk factor to predict the mortality in critically ill and injured patients who underwent emergent abdominal surgery. Moreover, the ROC curve for the DNI on POD1 confirmed that the optimal cut-off for predicting mortality was 7.1%, and the sensitivity (85.7%) and specificity (84.4%) were high with an AUC of 0.887. Mean DNI had a different pattern throughout the study period compared with other biomarkers such as WBC count and CRP. The mean DNI in survivors decreased after initial elevation whereas it continuously increased in non-survivors. Additionally, a recent study on DNI in patients with sepsis caused by peritonitis reported that DNI on POD3 was an independent risk factor for postoperative mortality, and the patterns of mean DNI were different between survivors and non-survivors. Moreover, the AUC for DNI was 0.88, and the optimal

**Table 3. Comparison between survivors and non-survivors.**

| | Survivor (n = 150) | Non-survivor (n = 19) | P-value |
|---|---|---|---|
| Age (year) | 53.9±16.8 | 53.6±19.6 | 0.944 |
| Sex (male) | 98 (65.3%) | 14 (73.7%) | 0.468 |
| Injury severity score (ISS) | 10 (1–50) | 25 (9–50) | <0.001 |
| Injury severity score (ISS) > 15 | 62 (41.3%) | 16 (84.2%) | <0.001 |
| Associated injury (AIS ≥3) | 45 (30%) | 12 (63.2%) | 0.004 |
| Shock | 35 (23.3%) | 12 (63.2%) | <0.001 |
| Initial WBC (/mm$^3$) | 12254±5670 | 10558±7514 | 0.353 |
| Initial DNI (%) | 0.45 (0–44.7) | 3.3 (0–52.8) | 0.053 |
| Initial CRP (mg/dL) | 0.29 (0–30.10) | 0.29 (0.29–23.80) | 0.215 |
| Initial Lactate (mmol/L) | 3.04±2.23 | 9.00±4.10 | <0.001 |
| GI perforation | 57 (38.0%) | 5 (26.3%) | 0.319 |
| Solid organ injury | 38 (25.3%) | 10 (52.6%) | 0.013 |
| SOFA score on ICU admission | 3 (0–15) | 9.5 (6–18) | <0.001 |
| Postoperative shock | 12 (8.0%) | 18 (94.7%) | <0.001 |
| Postoperative WBC (/mm$^3$) | 11096±5091 | 7161±5139 | 0.002 |
| Postoperative DNI (%) | 3.3 (0–41.2) | 9.0 (2.3–48.8) | 0.009 |
| Postoperative CRP (mg/dL) | 0.32 (0–26.40) | 0.29 (0–13.70) | 0.921 |
| Postoperative lactate (mmol/L) | 2.82±1.73 | 7.64±3.54 | <0.001 |
| POD1 WBC (/mm$^3$) (n = 163) | 9846±3620 | 7017±3949 | 0.005 |
| POD1 DNI (%) (n = 162) | 1.5 (0–35.9) | 17.0 (1.4–57.0) | 0.001 |
| POD1 CRP (mg/dL) (n = 153) | 10.40 (0–35.80) | 2.80 (0–17.30) | 0.014 |
| POD2 WBC (/mm$^3$) (n = 158) | 8695±3080 | 5561±3972 | 0.001 |
| POD2 DNI (%) (n = 157) | 0.8 (0–52.9) | 27.9 (0–62.1) | 0.004 |
| POD2 CRP (mg/dL) (n = 145) | 15.20 (0–35.10) | 7.32 (0–32.1) | 0.013 |
| ICU stay (day) | 4 (1–90) | 3 (1–58) | 0.450 |
| Duration of hospitalization (day) | 21 (2–697) | 3 (1–58) | <0.001 |

AIS, abbreviated injury scale; WBC, white blood cell; DNI, delta neutrophil index; CRP, C-reactive protein; SOFA, sequential organ failure assessment; POD, postoperative day; ICU, intensive care unit.

cut-off value was 7.8%, with a sensitivity of 77.3% and specificity of 95.9% [7]. This result is quite similar to that of our study. Indeed, in our hospital, it was possible to recognize the high probability of death in patients with elevated DNI level above 7.1% the day after surgery. DNI helped surgeons explain the patient's condition earlier to the caregiver and determine further evaluation and general ward transfer.

Previous studies about surgical and medical patients showed that DNI was a useful biomarker to predict disease severity or prognosis in patients with various infections or sepsis [7, 10, 16, 17]. However, studies about usefulness of DNI in patients who had tissue injury or hemorrhage due to trauma are limited [18]. Systematic inflammatory response syndrome (SIRS) is initiated within 30 minutes after severe injury, which is associated with an inflammatory response to hemorrhage or tissue damage rather than infections. Damage-associated molecular patterns (DAMPs) are released into the extracellular space by tissue damage that triggers an inflammatory response without infection. DAMPs activate the innate immune systems such as neutrophils, monocytes, and complements. This change can result in endothelial and organ damage and immunoparesis. As a result, a vicious cycle is triggered due to continuous inflammation and immune activation. In this process, neutrophils are activated initially by SIRS, but their bactericidal function is markedly impaired. This functional impairment is

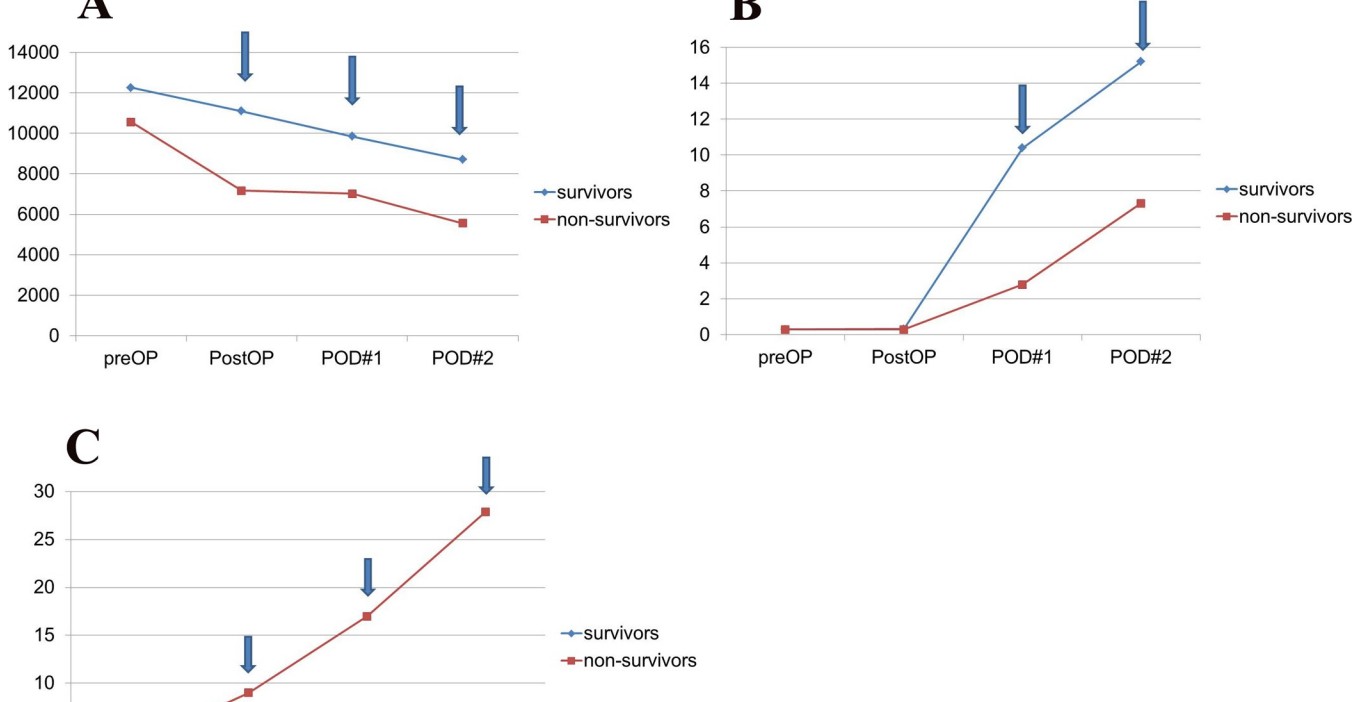

**Fig 2.** (A) White blood cell (WBC) count during the perioperative period. The mean WBC counts of survivors were significantly higher than that of the non-survivors on the immediate postoperative day, postoperative day 1, and postoperative day 2. (B) C-reactive protein (CRP) during the perioperative period. The mean CRP levels of the survivors were significantly higher than that of the non-survivors on postoperative day 1 and postoperative day 2. (C) Delta neutrophil index (DNI) during the perioperative period. The mean DNI of the non-survivors was significantly higher than that of the survivors on the immediate postoperative day, postoperative day 1, and postoperative day 2.

compensated by the release of immature banded neutrophils [19]. In the present study, DNI appears to be correlated with the severity of SIRS induced by tissue damage and hemorrhage in patients with severe trauma. In other words, increase in the value of DNI in non-survivors indicated severe SIRS, which may result in poor clinical outcomes as a result of progressive persistent inflammation, immunosuppression, and catabolism syndrome [20]. A recent study on trauma patients who were admitted in ICU reported that high DNI values at 12 and 24 hours from admission were strong independent predictors of multiple organ dysfunction

**Table 4. Independent risk factors for postoperative mortality.**

| Variable | Risk factors for mortality | |
|---|---|---|
| | Odd ratio (95% CI) | P-value |
| Initial shock | 0.096 (0.007–1.311) | 0.079 |
| Postoperative lactate level (mmol/L) | 1.926 (1.201–3.089) | **0.007** |
| SOFA score on ICU admission | 1.593 (1.160–2.187) | **0.004** |
| POD1 DNI (%) | 1.118 (1.028–1.215) | **0.009** |

Injury severity score>15, solid organ injury, combined injury, initial shock, postoperative lactate level, SOFA score on ICU admission, and DNI on POD1 were used as variables for multivariate analysis. CI, confidence interval; DNI, delta neutrophil index; SOFA, sequential organ failure assessment; POD1, postoperative day 1; ICU, intensive care unit.

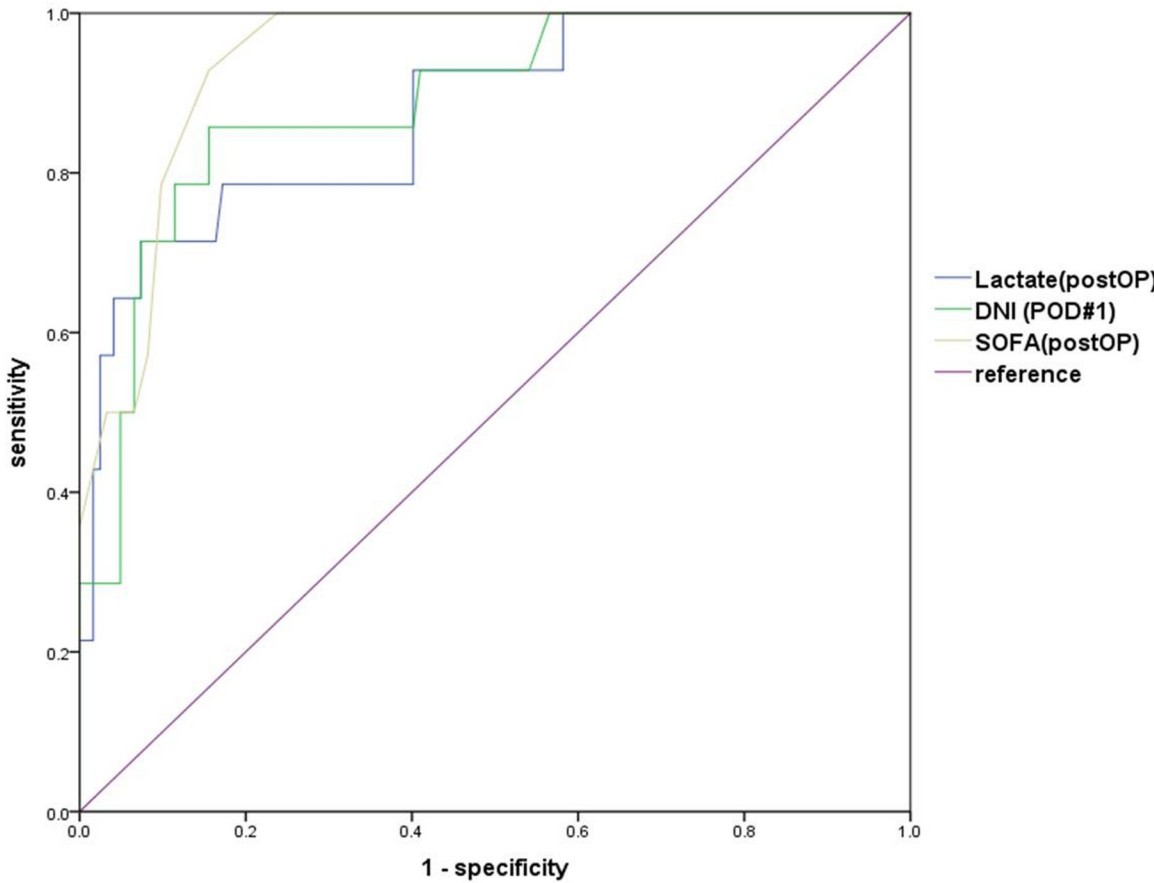

**Fig 3. Receiver operating characteristics (ROC) curves for the delta neutrophil index (DNI) (postoperative day 1, POD1), lactate (immediate postoperative, PostOP), and sequential organ failure assessment (SOFA) score (PostOP) between the survivors and non-survivors.**

syndrome (MODS). Increased DNI at 12 hours was a strong independent predictor of short-term mortality. In addition, the cut-off value of DNI at 12 hours to predict 30-day mortality was 5.3% in that study [18], which was similar to the result in our study.

In this study, we found that although 62 (36.7%) patients had GI perforation, there was no significant difference in the mortality rate according to GI perforation (patients with GI perforation 8.1% vs. patients without GI perforation 13.1%, p = 0.319). This result has several implications. First, most patients with abdominal injuries underwent early surgery if necessary, and therefore, many patients with GI perforation might not progress to septic shock. Second, there are causes other than infection that exacerbate the condition of a trauma patient. As described above, SIRS due to tissue damage and hemorrhage was a major deterioration factor in the early stage of major trauma and was closely associated with patient severity and prognosis. Severe SIRS seemed to cause MODS and eventually lead to death [19, 20].

Several studies suggested that WBC count was only a nonspecific indicator of stress such as infection, inflammation, tissue necrosis, and hemorrhage in injured patients [21, 22]. Another study on patients with abdominal injury showed that the WBC count within 24 hours after admission had limited diagnostic value for predicting hollow viscus injury [23]. Similarly, the present study showed that the WBC count had poor predictability in distinguishing survivors from non-survivors over time after surgery. Although there was a significant difference in the

CRP level between the two groups on POD1 and POD2, it is difficult to use CRP as a predictor of mortality, because the survivor group had higher CRP level than the non-survivor group, and CRP had a relatively slow changing pattern. A study performed in patients with multiple trauma showed that CRP increased easily and had very slow kinetics compared to other bio-markers [24]. In contrast, several studies have shown that initial lactate and lactate clearance are clinically useful in predicting mortality of trauma patients [25–27]. In our study, we found that postoperative lactate level was an independent risk factor for postoperative mortality and had a high predictability of mortality with an optimal cut-off level of 5.105 mmol/L (AUC 0.874, sensitivity 71.4%, and specificity: 92.6%). Taken together, DNI on POD1, postoperative SOFA score, and postoperative lactate may be used complementarily to predict patient death.

There are some limitations to our study. First, it is hard to confirm the usefulness of DNI, because of the small sample size and single institutional study. Second, data of the present study did not include serial levels of procalcitonin which is currently an important biomarker for diagnosis of infection and sepsis, because reimbursement for procalcitonin is limited within twice a week in the Korean national medical insurance. Third, there may be a selection bias in the present study, because of its retrospective nature. Despite these limitations, our study may be meaningful in that it is the first study to evaluate the usefulness of DNI to predict clinical outcome in abdominal trauma patients who underwent emergency surgery. In the future, large-scale prospective studies will be needed to confirm the results of our study.

## Supporting information

**S1 Table. Comparison of other variables between survivors and non-survivors.**
(DOCX)

**S1 Dataset.**
(XLSX)

## Author Contributions

**Conceptualization:** Kwangmin Kim, Hongjin Shim, Pil Young Jung, Young Un Choi, Keum Seok Bae, Ik Yong Kim, Ji Young Jang.

**Data curation:** Hui-Jae Bang, Kwangmin Kim, Seongyup Kim, Pil Young Jung, Young Un Choi, Ji Young Jang.

**Formal analysis:** Ji Young Jang.

**Investigation:** Hui-Jae Bang, Hongjin Shim, Ji Young Jang.

**Methodology:** Hongjin Shim, Keum Seok Bae, Ji Young Jang.

**Project administration:** Ji Young Jang.

**Software:** Ji Young Jang.

**Supervision:** Kwangmin Kim, Hongjin Shim, Seongyup Kim, Pil Young Jung, Keum Seok Bae, Ik Yong Kim, Ji Young Jang.

**Visualization:** Ji Young Jang.

**Writing – original draft:** Hui-Jae Bang, Ji Young Jang.

**Writing – review & editing:** Hui-Jae Bang, Kwangmin Kim, Hongjin Shim, Seongyup Kim, Pil Young Jung, Young Un Choi, Keum Seok Bae, Ik Yong Kim, Ji Young Jang.

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
