## [Decision Letter · Decision Letter 0]

10 Feb 2020

PONE-D-19-35229

Delta neutrophil index for predicting mortality in trauma patients who underwent emergent abdominal surgery: A case controlled study

PLOS ONE

Dear Dr. Jang,

Thank you for submitting your manuscript to PLOS ONE. After careful consideration, we feel that it has merit but does not fully meet PLOS ONE’s publication criteria as it currently stands. Therefore, we invite you to submit a revised version of the manuscript that addresses the points raised during the review process.

We would appreciate receiving your revised manuscript by Mar 26 2020 11:59PM. To enhance the reproducibility of your results, we recommend that if applicable you deposit your laboratory protocols in protocols.io, where a protocol can be assigned its own identifier (DOI) such that it can be cited independently in the future. For instructions see: http://journals.plos.org/plosone/s/submission-guidelines#loc-laboratory-protocols

We look forward to receiving your revised manuscript.

Kind regards,

Itamar Ashkenazi

Academic Editor

PLOS ONE

Journal Requirements:

"This research received no specific grants from funding agencies in the public, commercial, or not-for-profit sectors."

We note that one or more of the authors are employed by a commercial company: Saidabad Clinic

Reviewers' comments:

Reviewer's Responses to Questions

**Comments to the Author**

1. Is the manuscript technically sound, and do the data support the conclusions?

Reviewer #1: Yes

Reviewer #2: Yes

2. Has the statistical analysis been performed appropriately and rigorously? 

Reviewer #1: Yes

Reviewer #2: Yes

3. Have the authors made all data underlying the findings in their manuscript fully available?

Reviewer #1: Yes

Reviewer #2: Yes

4. Is the manuscript presented in an intelligible fashion and written in standard English?

Reviewer #1: Yes

Reviewer #2: Yes

5. Review Comments to the Author

Reviewer #1: The article by Jang explores mortality prediction in trauma patients undergoing emergent abdominal surgery.

The DNI is the immature granulocyte fraction determined by subtracting the fraction of mature PMN leukocytes from the sum of myeloperoxidase-reactive cells and reflects the number of immature neutrophils. It has been used in sepsis and several other conditions.

Several questions: 1. What were the inclusion/exclusion criteria?

2. Was the study exploring in hospital deaths only or was there an attempt to examine 28 days mortality or longer? How does the death distribution look like?

3. In Table 1&3, is this the mean?

4. Can a predictor based on DNI difference and the other idependent variable identified and POD be presented in an aggregate score so that it would have clinical utility for the practicing surgeon?

5. Could the author add a sentence about the scientific rationale why increased DNI is associated with mortality?

Reviewer #2: The manuscript looks interesting, well written and intelligible.

The methodology of the study looks appropriate and rigorous.

The work seems to add something new to the field of trauma research.

My recommendation is to accept.

6. PLOS authors have the option to publish the peer review history of their article (what does this mean?). If published, this will include your full peer review and any attached files.

Reviewer #1: No

Reviewer #2: Yes: Roberto Faccincani

---

## [Author Response · Author response to Decision Letter 0]

13 Feb 2020

Reviewer #1: The article by Jang explores mortality prediction in trauma patients undergoing emergent abdominal surgery.

The DNI is the immature granulocyte fraction determined by subtracting the fraction of mature PMN leukocytes from the sum of myeloperoxidase-reactive cells and reflects the number of immature neutrophils. It has been used in sepsis and several other conditions.

Several questions: 1. What were the inclusion/exclusion criteria?

To clarify the inclusion and exclusion criteria, we added a patient flow chart to Figure 1. 

2. Was the study exploring in hospital deaths only or was there an attempt to examine 28 days mortality or longer? How does the death distribution look like?

Twelve (63.2%) patients died within 7 days, and 4 (21.1%) died between 7 and 28 days. The other one died of sepsis after 28 days. 

3. In Table 1&3, is this the mean?

As we described in ‘statistical analysis’ section of patients and methods, continuous variables were mainly presented as the mean±standard deviation. When no following a normal distribution, it was expressed as median(range). 

4. Can a predictor based on DNI difference and the other idependent variable identified and POD be presented in an aggregate score so that it would have clinical utility for the practicing surgeon?

In multivariated analysis, POD1 DNI, postoperative lactate, and postoperative SOFA score were identified as independent risk factors related to mortality. Authors confirmed the AUC using the ROC curve of these variables, and confirmed that the cut-off value of POD1 DNI was the criterion having the highest sensitivity and specificity at 7.1%. The postoperative lactate and SOFA scores were also analyzed in the same way, indicating cut-off levels of 5.1 and 6.5. 

Indeed, in our hospital, it was possible to recognize the high probability of death in patients with elevated DNI levels above 7.1% the day after surgery. Doctors were able to explain the patient’s condition earlier to the caregiver and helped determine further evaluation and general ward transfer. However, we did not calculate a score that combines these three scores. 

5. Could the author add a sentence about the scientific rationale why increased DNI is associated with mortality?

DNI can be seen as a value representing the proportion of immature granulocytes. As SIRS caused by tissue damage increases the release of damage-associated molecular patterns (DAMPs), SIRS becomes more severe and the multiple organ failure of the patient worsens. In addition, when neutrophils are functionally impaired by SIRS, the release of immature banded neutrophils increases to compensate for this, which is indicated by an increase in DNI. This is described in the discussion section (243-249th line). 

Reviewer #2: The manuscript looks interesting, well written and intelligible.

The methodology of the study looks appropriate and rigorous.

The work seems to add something new to the field of trauma research.

My recommendation is to accept.

---

## [Editor Report · Decision Letter 1]

17 Feb 2020

PONE-D-19-35229R1

Delta neutrophil index for predicting mortality in trauma patients who underwent emergent abdominal surgery: A case controlled study

PLOS ONE

Dear Dr. Jang,

I am returning the revised manuscript since I need some clarifications.

You respond to each of the first reviewer's questions but except in one case I do not see in your response letter a description of what changes did you make within the manuscript. Associated to this issue, when I examined the marked copy, I did not always understand why certain places were highlighted.

I suggest the following. Under each of the reviewer's questions introduce two subheadings: authors' response, desciption of changes made. Thus for each of the questions we will have your answer followed by an explanation if changes were made in the manuscript and their location. If no changes were made, just write that no changes were made.

Thank you,

Itamar Ashkenazi M.D.

Academic Editor

---

## [Author Response · Author response to Decision Letter 1]

18 Feb 2020

Dear Academic Editor 

As you mentioned, we answered reviewer’s questions point-by-point and clearly stated whether the manuscript was corrected or not. 

Also we highlighted again in the only changed points of the manuscript. 

Thank you for your comments. 

Ji Young Jang M.D. 

Review Comments to the Author

Reviewer #1: The article by Jang explores mortality prediction in trauma patients undergoing emergent abdominal surgery.

The DNI is the immature granulocyte fraction determined by subtracting the fraction of mature PMN leukocytes from the sum of myeloperoxidase-reactive cells and reflects the number of immature neutrophils. It has been used in sepsis and several other conditions.

Several questions: 

1. What were the inclusion/exclusion criteria?

To clarify the inclusion and exclusion criteria

 We added a patient flow chart as Figure 1(line 87-88).

2. Was the study exploring in hospital deaths only or was there an attempt to examine 28 days mortality or longer? How does the death distribution look like?

Thirteen (68.4%) patients died within 7 days, and 5 (26.3%) died between 7 and 28 days. The other one died of sepsis after 28 days. 

 We added this contents to the Results section (line 140-141). 

3. In Table 1&3, is this the mean?

As we described in ‘statistical analysis’ section of patients and methods, continuous variables were mainly presented as the mean±standard deviation. When no following a normal distribution, it was expressed as median(range). 

4. Can a predictor based on DNI difference and the other idependent variable identified and POD be presented in an aggregate score so that it would have clinical utility for the practicing surgeon?

In multivariated analysis, POD1 DNI, postoperative lactate, and postoperative SOFA score were identified as independent risk factors related to mortality. Authors confirmed the AUC using the ROC curve of these variables, and confirmed that the cut-off value of POD1 DNI was the criterion having the highest sensitivity and specificity at 7.1%. The postoperative lactate and SOFA scores were also analyzed in the same way, indicating cut-off levels of 5.1 and 6.5. 

Indeed, in our hospital, it was possible to recognize the high probability of death in patients with elevated DNI levels above 7.1% the day after surgery. DNI helped surgeons explain the patient’s condition early to the caregiver and determine further evaluation and general ward transfer. However, we did not calculate a score that combines these three scores. 

Authors added this content to the discussion section (line 232-235).

5. Could the author add a sentence about the scientific rationale why increased DNI is associated with mortality?

DNI can be seen as a value representing the proportion of immature granulocytes. As SIRS caused by tissue damage increases the release of damage-associated molecular patterns (DAMPs), SIRS becomes more severe and the multiple organ failure of the patient worsens. In addition, when neutrophils are functionally impaired by SIRS, the release of immature banded neutrophils increases to compensate for this, which is indicated by an increase in DNI. 

Since this content is already described in the discussion section, authors did not make any additional descriptions (239-248th line). 

Reviewer #2: The manuscript looks interesting, well written and intelligible.

The methodology of the study looks appropriate and rigorous.

The work seems to add something new to the field of trauma research.

My recommendation is to accept.

---

## [Editor Report · Decision Letter 2]

20 Feb 2020

PONE-D-19-35229R2

Delta neutrophil index for predicting mortality in trauma patients who underwent emergent abdominal surgery: A case controlled study

PLOS ONE

Dear Dr. Jang,

Thank you for submitting your manuscript to PLOS ONE. Thank you also for clarifying the reviewers' comments and making the appropriate changes. I want to aplogize but there is one more issue that needs to be dealt with before considerning this manuscript suitable for publication.

ISS and SOFA are scores based on several components.  Considering these as continuous variables is not appropriate.  There is no ISS 15 for example.  ISS 16 is not twice as bad as ISS 8...

I recommend either presenting these as medians and ranges, or presenting proportion of patients above a certain threshold. For example, proportion of patients with ISS of 16 or higher. 

Please introduce these before I hand in the final decision.  

We would appreciate receiving your revised manuscript by Apr 05 2020 11:59PM. To enhance the reproducibility of your results, we recommend that if applicable you deposit your laboratory protocols in protocols.io, where a protocol can be assigned its own identifier (DOI) such that it can be cited independently in the future. For instructions see: http://journals.plos.org/plosone/s/submission-guidelines#loc-laboratory-protocols

We look forward to receiving your revised manuscript.

Kind regards,

Itamar Ashkenazi

Academic Editor

PLOS ONE

---

## [Author Response · Author response to Decision Letter 2]

20 Feb 2020

PONE-D-19-35229R2

Delta neutrophil index for predicting mortality in trauma patients who underwent emergent abdominal surgery: A case controlled study

PLOS ONE

Dear Dr. Jang,

Thank you for submitting your manuscript to PLOS ONE. Thank you also for clarifying the reviewers' comments and making the appropriate changes. I want to aplogize but there is one more issue that needs to be dealt with before considerning this manuscript suitable for publication.

ISS and SOFA are scores based on several components. Considering these as continuous variables is not appropriate. There is no ISS 15 for example. ISS 16 is not twice as bad as ISS 8...

I recommend either presenting these as medians and ranges, or presenting proportion of patients above a certain threshold. For example, proportion of patients with ISS of 16 or higher. 

Please introduce these before I hand in the final decision. 

 Thank you for your important comment. 

The authors discussed the reviewer’s opinion and decided to present proportion (ISS>15) and median (range) for the ISS. However, the SOFA score on ICU admission was linearly correlated with the mortality rate of patients, we decided to keep it as a continuous variable. Since the standard deviation of SOFA score is relatively large, we modified it to median (range). 

The multivariate analysis was re-run using the modified ISS>15 and confirmed no change in the main results. In addition, tables and manuscript were revised. 

Reference

Ferreira et al. Serial evaluation of the SOFA score to predict outcome in critically ill patients. JAMA.2001 Oct 10;286(14):1754-8 

Thank you gain. 

Ji Young Jang M.D.

---

## [Editor Report · Decision Letter 3]

24 Feb 2020

Delta neutrophil index for predicting mortality in trauma patients who underwent emergent abdominal surgery: A case controlled study

PONE-D-19-35229R3

Dear Dr. Jang,

We are pleased to inform you that your manuscript has been judged scientifically suitable for publication and will be formally accepted for publication once it complies with all outstanding technical requirements.

With kind regards,

Itamar Ashkenazi

Academic Editor

PLOS ONE
---

## [Editor Report · Acceptance letter]

6 Mar 2020

PONE-D-19-35229R3 

Delta neutrophil index for predicting mortality in trauma patients who underwent emergent abdominal surgery: A case controlled study 

Dear Dr. Jang:

I am pleased to inform you that your manuscript has been deemed suitable for publication in PLOS ONE. Congratulations! Your manuscript is now with our production department. 

With kind regards,

on behalf of

Dr. Itamar Ashkenazi 

Academic Editor

PLOS ONE